

# Detection of fake face images using lightweight convolutional neural networks with stacking ensemble learning method

Emre Şafak[1,2] and Necaattin Barışçı[2]

[1] R&D Technology and Innovation Department, HAVELSAN, Ankara, Türkiye
[2] Department of Computer Engineering, Gazi University Ankara, Ankara, Türkiye

## ABSTRACT

Images and videos containing fake faces are the most common type of digital manipulation. Such content can lead to negative consequences by spreading false information. The use of machine learning algorithms to produce fake face images has made it challenging to distinguish between genuine and fake content. Face manipulations are categorized into four basic groups: entire face synthesis, face identity manipulation (deepfake), facial attribute manipulation and facial expression manipulation. The study utilized lightweight convolutional neural networks to detect fake face images generated by using entire face synthesis and generative adversarial networks. The dataset used in the training process includes 70,000 real images in the FFHQ dataset and 70,000 fake images produced with StyleGAN2 using the FFHQ dataset. 80% of the dataset was used for training and 20% for testing. Initially, the MobileNet, MobileNetV2, EfficientNetB0, and NASNetMobile convolutional neural networks were trained separately for the training process. In the training, the models were pre-trained on ImageNet and reused with transfer learning. As a result of the first trainings EfficientNetB0 algorithm reached the highest accuracy of 93.64%. The EfficientNetB0 algorithm was revised to increase its accuracy rate by adding two dense layers (256 neurons) with ReLU activation, two dropout layers, one flattening layer, one dense layer (128 neurons) with ReLU activation function, and a softmax activation function used for the classification dense layer with two nodes. As a result of this process accuracy rate of 95.48% was achieved with EfficientNetB0 algorithm. Finally, the model that achieved 95.48% accuracy was used to train MobileNet and MobileNetV2 models together using the stacking ensemble learning method, resulting in the highest accuracy rate of 96.44%.

# INTRODUCTION

Since the widespread use of the internet fake images and videos have been produced to deceive or entertain people. Artificial intelligence algorithms have been used in recent years for manipulations that were initially made using image and video editing programs. Deep learning based methods such as automatic encoders and generative adversarial networks are often used to generate fake image and video using artificial intelligence

Corresponding author
Emre Şafak, emresfk2@gmail.com

(*Pashine et al., 2021*). Face image manipulation is the most common type of fake image produced. Most fake face images are intended to discredit or scam people. With fake face images people can be shown where they are never been and can be made to speak with words they did not say. Fake face manipulations are of four types: identity manipulation (deepfake), attribute manipulation, expression manipulation, and entire face synthesis (*Wang et al., 2021*).

Identity manipulation is the replacement of the face image of the person in a video or image with another person. The selected face is replaced with another identified face in the video or image. Computer graphics-based methods (face swap) and deep learning (deepfake) algorithms are generally used for identity manipulation. The deepfake based approach relies on two autoencoders trained to reconstruct the training images of the source and target face. A face detector is used to crop and align images. The source face's trained encoder and decoder are applied to the target face to create a fake image andvideo. The autoencoder output is then aligned with the rest of the image using the Poisson image editing method (*Tolosana et al., 2020*).

Attribute manipulation involves changing physical features of the face, such as hair color, skin color, age, gender, and eyes. For this method of manipulation, generative adversarial networks often called StarGAN are used. StarGAN (*Choi et al., 2018*) consists of a discriminator (D) and a generator (G). The discriminator tries to guess whether an input image is fake or real and classifies the real image according to its corresponding domain. The generator takes both the image and the target domain label as input and creates a fake image. The generator tries to reconstruct the original image from the fake image supplied with the original domain label provided by the discriminator. Finally, the generator tries to produce images that are indistinguishable from real images and can be classified as target areas by the discriminator. The widely used FaceApp mobile application is one of the sample applications that allows the user to manipulate attributes. Users can use this technology to try various products (cosmetics, makeup, hairstyles, *etc.*,) in a virtual environment (*Wang et al., 2021*).

Expression manipulation involves changing the expressions or emotions displayed on a person's face. Facial expressions (happy, excited, confused, angry, *etc.*) can be changed with expression manipulation also known as face animation. With this method images can be animated and used as desired. For example, with expression manipulation a person can make a speech in a place where they have never been. For this reason, expression manipulation can have serious consequences (*Wang et al., 2021*). Face2Face (*Thies et al., 2016*) and Neural Nextures (*Thies, Zollhöfer & Nießner, 2019*) methods are used for expression manipulation.

Entire face synthesis is the generation of nonexistent face images using generative adversarial networks. Strong generative adversarial networks such as StyleGAN (*Karras & Hellsten, 2021a*) are generally used for entire face synthesis manipulation. StyleGAN is a generative adversarial network that enables the generation of realistic fake face images developed by NVIDIA researchers. Face images can be produced with a high level of realism with entire face synthesis. Entire face synthesis can be used in various applications such as gaming, 3D modeling, media, *etc*. While it can benefit many industries, it can also

be used to create very realistic fake profiles on social networks to spread disinformation (*Tolosana et al., 2020*).

With the use of artificial intelligence algorithms in face manipulations, it has become very difficult to distinguish between real content and fake content. Due to the increase in the number and quality of fake face images, studies have started to be carried out in order to detect them. Most of the high-performance fake face detection methods are based on deep learning. Usually, convolutional neural network or recurrent neural network techniques are used. In addition to being able to detect fake faces with high accuracy, resources should be used effectively. In traditional approaches data collected from data sources are processed on central servers. With the increasing amount of data and the internet of things connecting billions of devices to the internet, the current processing capacities and bandwidths will not be enough (*Şafak et al., 2021*). For this reason, the fake face detection model to be developed should be able to work with maximum performance on mobile devices with low processing power instead of a central processing server. In this study, fake face images produced by generative adversarial networks were detected by using lightweight convolutional neural networks such as MobileNet, MobileNetV2, EfficientNetB0 and NASNetMobile that can work on mobile devices. A dataset containing 70,000 real and 70,000 fake images was used for the training process. EfficientNetB0 algorithm has reached the highest accuracy rate as a result of the training processes. In order to increase the accuracy rate, the highest accuracy rate of 96.44% was achieved when EfficientNetB0, MobileNet and MobileNetV2 were trained together with the stacking ensemble learning method.

Previous studies have generally used complex and costly models for fake face detection. Previous studies where lightweight convolutional neural networks were used, high accuracy values could not be achieved. Therefore, in this study, lightweight convolutional neural networks requiring lower computational cost were used. The main contributions of the article are as follows:

- Models that can run on mobile devices were trained and tested on the FFHQ dataset.
- MobileNet, MobileNetV2, EfficientNetB0, and NASNetMobile lightweight convolutional neural networks were trained separately. Models pre-trained on ImageNet were reused with transfer learning. In the tests conducted, the EfficientNetB0 algorithm achieved the highest accuracy rate of 93.64%.
- By fine-tuning the EfficientNetB0 model that achieved the highest accuracy rate, two dense layers with ReLU activation function (256 neurons), two dropout layers, a flatten layer, one dense layer with ReLU activation function (128 neurons), and two dense layers with softmax activation function for classification were added. The EfficientNetB0 model achieved an accuracy rate of 95.48%.
- Models trained separately were re-trained using transfer learning to increase the accuracy rate. By training the MobileNet and MobileNetV2 models together with the EfficientNetB0 model using the stacking ensemble learning method, the highest accuracy rate of 96.44% was achieved.

- When the proposed method was trained and tested on the CelebA-HQ dataset, an accuracy rate of 94.52% was achieved. Thus, the generalizability of the proposed method to different datasets has been demonstrated.

When the proposed method was run on a mobile device, the latency was 0.171 s. This demonstrated that the proposed method can be used in real-world scenarios.

In this article, literature research, materials and methods used, research findings and results are explained. In the second chapter, literature review is presented. In the third chapter, the method and data set used in this study are explained. In the fourth chapter, the experimental results of the proposed method and its comparison with other studies are given.

## LITERATURE REVIEW

A new method based on monitoring the neuron behavior of the layers of the convolutional neural network model is proposed for the detection of fake face images. Studies show that deep learning systems that monitor neuron coverage and behavior give successful results against adversarial attacks made on the model. In this study, it was observed that monitoring neuron behaviors gave similar results. Better detection of spurious patterns was achieved with the application of neuron activation in convolutional neural network layers. Experimental results showed that the proposed method gave better results in detecting identity manipulation, attribute manipulation, expression manipulation and entire face synthesis manipulation type. In the proposed method, 91.9% accuracy rate was achieved in the fake face image dataset produced with StyleGAN2 using the FFHQ dataset in the entire face synthesis manipulation method (*Wang et al., 2021*).

FisherFace and Local Binary Pattern Histogram (LBPH) algorithms were used together for the detection of fake face images. FisherFace was used to reduce the size of the face field, while LBPH was used to classify the face images. FisherFace is basically based on Fisher's Linear Discriminant Analysis (FLDA) method. The most important advantage of the FisherFace algorithm is that it works faster than other existing algorithms and has a low error rate. LBPH recognizes the image with less computational complexity by extracting the features of the face images. As fake face images dataset FFHQ, 100K-Faces, DFFD, CASIA-WebFace datasets were used separately and compared. The proposed method reached 94.92% accuracy on the FFHQ dataset (*Suganthi et al., 2022*).

A fine-tuned neural network architecture based on dual attention is proposed to detect fake face images. In the proposed method, the pre-trained model is integrated with a fine-tuning converter, MobileNet block V3 and channel attention module to increase performance and robustness. The fine-tuning converter consists of its own attention module and a subsampling layer. Moreover, the classifier can be easily integrated with other convolutional neural networks, while using less data for fine-tuning to increase performance and robustness. The channel attention module is used to capture the feature map of fake images. MobileNetV3 convolutional neural network has been used to classify images. The FaceForensics++ dataset and various generative adversarial network (GAN) generated datasets were used to test the proposed method. The effectiveness of the model

was checked on eight different generative adversarial networks. While the proposed method reached 81.43% accuracy on the dataset produced with StyleGAN, it reached 83.64% accuracy on the dataset produced with StyleGAN2 (*Bang & Woo, 2021*).

A method proposes using only the eyes to detect fake face images. Fake facial images were detected only by considering the inconsistent corneal specular between the two eyes. Normally, corneal highlights of real human face images are related to each other, but this relationship cannot be achieved in fake face images. In a real human face image, the front position of both eyes is in the same direction as the camera. Moreover, the eyes are away from the light source and all light sources in the environment are visible to both eyes. While the FFHQ dataset was used for real face images, the dataset created by the StyleGAN2 method was used for fake face images. As a result of the proposed method, the area under the curve (AUC) reached 94% (*Hu, Li & Lyu, 2020*).

A method proposes using irregular pupil shapes to detect fake face images. In real eyes, the pupils are circular or elliptical, whereas the pupils formed by the contentious generative meshes are irregular. The proposed method first uses a face detector to identify the face in the input image. U-Net-based model developed with EfficientNet-B5 is used to detect the boundaries of the pupil. Then, the widely used BIoU for distance measure was used to calculate the alignment and irregularity of the pupil with the ellipse. According to this calculation result, it can be determined that the image is fake or real. In this study, the FFHQ dataset was used for real face images, and the dataset created using the StyleGAN2 method and consisting of 1,600 images was used for fake face images. As a result of the proposed method, the AUC reached 91% (*Guo et al., 2022*).

A summary of the studies on fake face detection is presented in Table 1.

In Table 1, among the studies conducted using the FFHQ dataset and StyleGAN2, the study employing Fisher's Linear Discriminant Analysis method achieved the highest accuracy rate of 94.92% (*Suganthi et al., 2022*). Another study, which considered inconsistent corneal specularities between the two eyes without using deep learning, achieved an AUC value of 94% (*Hu, Li & Lyu, 2020*). In a method based on monitoring neuron behaviors in the layers of a convolutional neural network model for detecting fake face images, an accuracy rate of 91.9% was attained (*Wang et al., 2021*). Additionally, considering iris patterns, another study using the EfficientNet-B5 algorithm reached an accuracy rate of 91% (*Guo et al., 2022*). Among the examined studies, the study utilizing the MobileNetV3 algorithm achieved the lowest accuracy rate of 83.64% (*Bang & Woo, 2021*). While most of the examined studies employed deep learning methods, StyleGAN2 was preferred for generating fake face images. The study using the MobileNetV3 lightweight convolutional neural network model achieved a lower accuracy rate compared to other studies. There are no lightweight convolutional neural network models capable of achieving high accuracy. Therefore, in this study, fake faces generated with both lightweight convolutional neural network models and StyleGAN2 were used.

## MATERIALS AND METHODS

Studies in the literature deep learning algorithms that provide higher accuracy are used instead of classical image processing techniques in detecting fake face images. In addition

**Table 1 Studies on fake face detection.**

| Study | Method | Algorithm | Feature | Dataset | Performance metric | Accuracy rate |
|---|---|---|---|---|---|---|
| *Wang et al. (2021)* | Deep learning | New model based on neuron monitoring | Fake face detection | FFHQ and StyleGAN2 | Accuracy | 91.9% |
| *Suganthi et al. (2022)* | Deep learning | FisherFace | Fake face detection | FFHQ | Accuracy | 94.92% |
| *Bang & Woo (2021)* | Deep learning | MobileNetV3 | Fake face detection | FFHQ and StyleGAN2 | Accuracy | 83.64% |
| *Hu, Li & Lyu (2020)* | Image processing | Dlib library | Fake face detection | FFHQ and StyleGAN2 | AUC | 94% |
| *Guo et al. (2022)* | Deep learning | EfficientNet-B5 | Fake face detection | FFHQ and StyleGAN2 | AUC | 91% |

to being able to detect fake faces with high accuracy, resources must be used efficiently. Connecting billions of devices to the Internet of Things (IoT) will make it increasingly difficult to perform transactions on centralized servers. For this, the fake face detection model to be developed should be able to work on mobile devices. For this reason, lightweight convolutional neural network models that can work at maximum performance on mobile devices were used in this study. The flowchart of the study is shown in Fig. 1.

Figure 1 shows the work done to develop the fake face detection model. A dataset containing 70,000 real and 70,000 fake images was used to train the fake face detection model. A total of 112,000 images in the dataset were used for training and 28,000 images for testing. MobileNet, MobileNetV2, EfficientNetB0 and NASNetMobile lightweight convolutional neural networks are used for the training. Firstly, MobileNet, MobileNetV2, EfficientNetB0, and NASNetMobile convolutional neural networks are trained separately for the training. In the training process, the models were pre-trained on ImageNet and reused with transfer learning. As a result of the first trainings, the EfficientNetB0 algorithm reached the highest accuracy rate of 93.64%. To further increase the accuracy of the EfficientNetB0 algorithm, two dense layers (each with 256 neurons) with ReLU activation function, two dropout layers, one smoothing layer, one dense layer (with 128 neurons) with ReLU activation function, and a dense layer with two nodes and softmax activation function were added for classification. As a result of this process, an accuracy rate of 95.48% has been achieved. Finally, with the model that reached 95.48% accuracy, MobileNet and MobileNetV2 models were trained together with the stacking ensemble learning method and the highest accuracy rate was achieved with 96.44%. The study of the convolutional neural network architecture that enables fake face detection is presented in Fig. 2.

As seen in Fig. 2, the feature map is created by passing the acquired image through the convolutional neural network layers. Convolutional neural networks were first developed

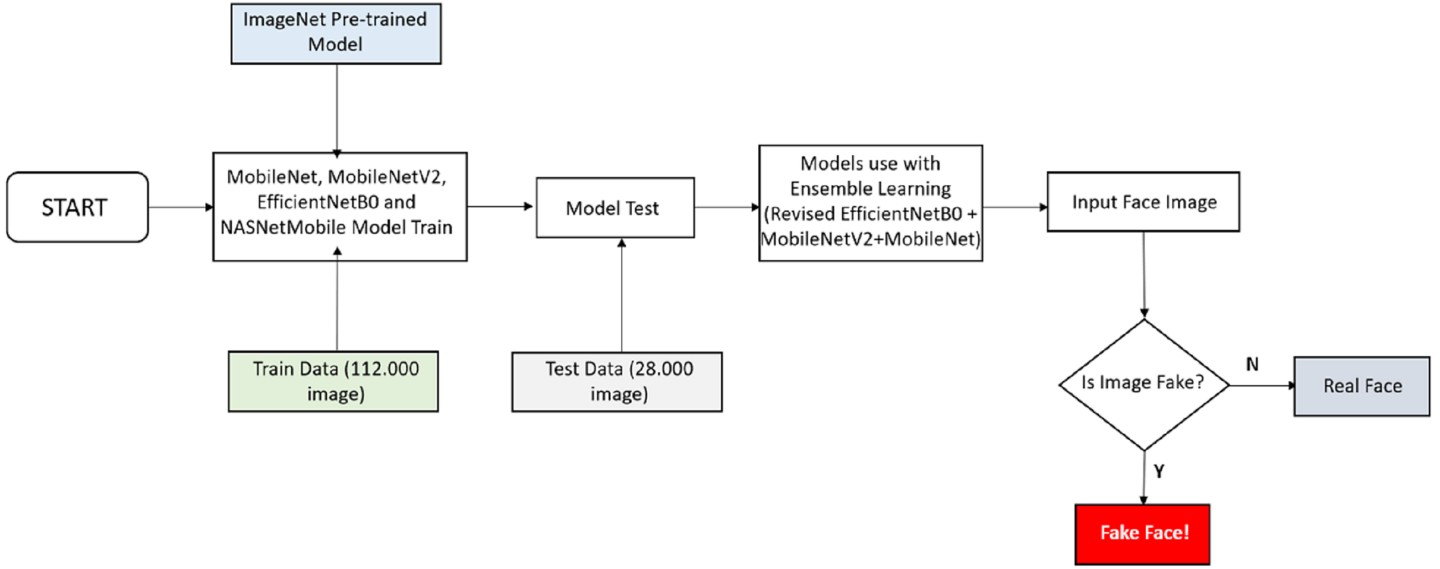

**Figure 1 Flowchart of the study.**

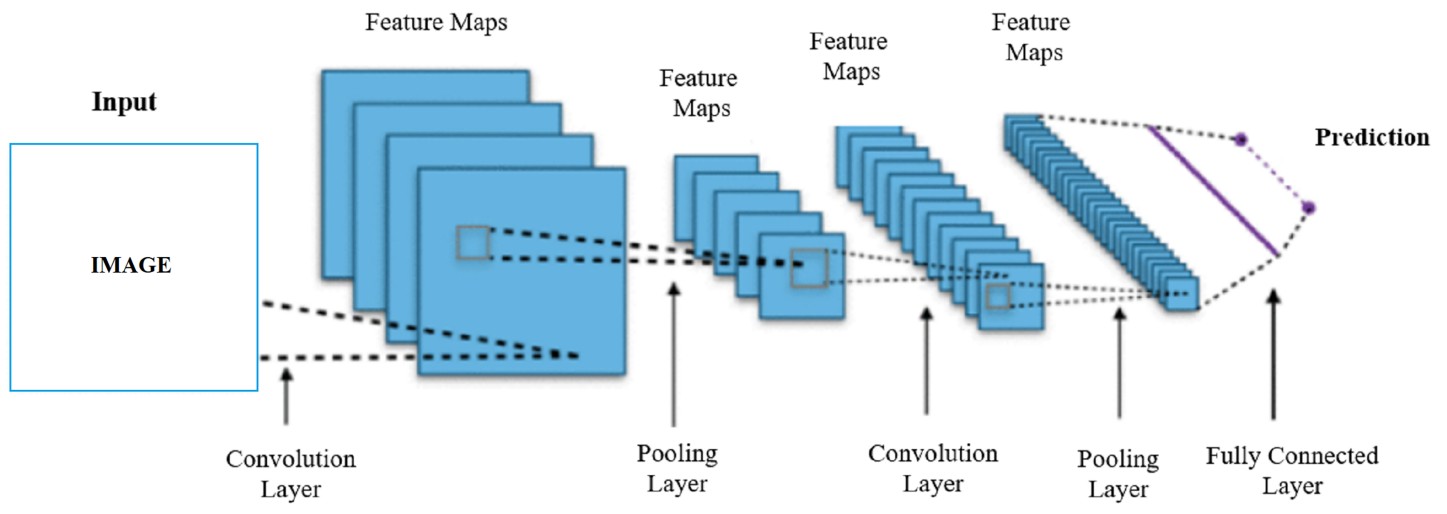

**Figure 2 Architecture of convolutional neural network.**

and used in 1980. Convolutional neural networks are artificial neural networks developed for deep learning that learn directly from data and eliminate the need for manual feature extraction. Convolutional neural networks perform the learning process by taking an input image and assigning learnable weights to various regions. Convolutional neural networks have proven themselves by giving very successful results especially in image recognition and classification tasks. Convolutional neural networks are also used in recognizing and classifying non-image data such as audio, time series, signal, and text data. A convolutional neural network developed for a particular task can be reused for another task.

Convolutional neural networks consist of many hidden layers between an input and output layer. The most commonly used hidden layers are convolution, ReLU, pooling and fully connected layer (*Şafak & Barışçı, 2018*). The convolution layer enables the detection of image features using convolution filters. The convolution layer first multiplies the convolution filters with the inputs from the neural network. During this multiplication process, the filter is passed over the image multiple times, by applying it from right to left and top to bottom, to cover the entire image. These products are then added together to create convolutional maps (*Albawi, Mohammed & Al-Zawi, 2017*). The ReLU layer introduces nonlinearity to the network by equating negative values to zero and keeping positive values. In this way, it enables faster and more effective training (*Şafak et al., 2022*). The pooling layer keeps only the most important information reducing the number of parameters in the input. The pooling layer helps prevent overfitting, reduce complexity, and increase efficiency by reducing the number of calculations and parameters in the network. A filter with non-trainable parameters is applied to the input, and based on the type of pooling, the filter generates an output array. There are two main types of pooling, maximum and average pooling. In the maximum pooling method as the filter moves through the input, it selects the pixel with the maximum value to be sent to the output array. In the mean pooling method, as the filter moves across the input, it calculates the average value within the receiving field to send to the output array (*İnik & Ülker, 2017*). The fully connected layer performs the classification function based on the attributes from the previous layers. The fully connected layer usually generates a probability between 0 and 1, using the softmax activation function, to properly classify the inputs (*Khan et al., 2020*).

The model classifies the input image as Fake Face or Real Face according to the feature map after passing through all the relevant convolutional neural network layers.

A dataset of 140,000 images was used to train the convolutional neural network models. The dataset used includes 70,000 real face images in the FFHQ dataset and 70,000 fake face images produced with StyleGAN2 (*Karras & Hellsten, 2021b*). The FFHQ dataset is a dataset consisting of high-resolution images of age and ethnicity diversity prepared for generative adversarial networks. 6.0% of the images in the dataset belong to the United States, 1.6% to the United Kingdom, 0.7% to Canada, 0.6% to Spain, 0.5% to Taiwan, and 5.7% to other known countries. The remaining 85% are of unknown origin (*Karras & Hellsten, 2022*). StyleGAN2 is very successful in data-driven unconditional generative modeling (*Karras et al., 2020*). Therefore fake faces created with StyleGAN2 are very realistic and challenging. While 80% of the images in the dataset were used for training and validation, 20% were used for testing (*Rácz, Bajusz & Héberger, 2021*).

The hyperparameters in Table 2 were used to train the model. The proposed model achieved its highest accuracy with the hyperparameters listed in Table 2.

The epoch value shows how many times the dataset is trained in the proposed convolutional neural network. In this study, the maximum accuracy was achieved when the dataset was trained on the proposed convolutional neural network 15 epochs. Since the number of datasets is high, the number of steps per epoch is set to 64 and the dataset is trained by taking 64 part at each step. Verbosity value is set to one in order to display the stage of model training. The learning rate which is the update rate of the learned weights in

**Table 2 Hyperparameters used in training.**

| Hyperparameters | Value |
|---|---|
| Epoch | 15 |
| Steps_per_epoch | 64 |
| Verbosity | 1 |
| Learning rate | 0.001 |
| Loss_function | Binary_crossentropy |
| Optimizer | Adam |

the model training, is set to 0.001. Binary cross entropy function is used because binary classification is made in order to calculate the loss between the model's prediction and the true value. Adam optimization algorithm was used to update the determined learning rate according to different parameters. The Adam optimization algorithm is preferred because it is computationally efficient, requires low memory requirements, and the dataset is suitable for large problems (*Kingma & Ba, 2017*). The model was trained using a computer with an Intel i7-12650H CPU, 32GB RAM, and NVIDIA GeForce RTX4090 16GB GPU. The training process took approximately 92 h.

MobileNet, MobileNetV2, EfficientNetB0, and NASNetMobile convolutional neural network models used for fake face detection in this study. MobileNet is an efficient and not very computationally intensive convolutional neural network developed for mobile and embedded applications. MobileNet 28 layers consist of 4.2 million parameters. MobileNet uses deeply separable convolutions to build lightweight convolutional neural networks. The number of parameters is significantly reduced by using deeply separable convolutions in the MobileNet. Deep separable convolution consists of two layers: depth convolution and point convolution. Deep convolution is used to apply a single filter to each input. Since deep convolution is only used to filter the input channel, it cannot combine these filters to generate new features. Point convolution is a $1 \times 1$ convolution that computes a linear combination of the in-depth convolution output. MobileNet requires much less computational power to run or implement transfer learning. MobileNet is also best suited for web browsers, as browsers have limitations on computing, graphics processing and storage (*Howard et al., 2017*). MobileNet architecture is presented in Fig. 3.

MobileNetV2 is a convolutional neural network that can work efficiently on mobile and embedded devices and aims to give better results in terms of performance. MobileNetV2 convolutional neural network consists of 53 layers and 3.4 million parameters. MobileNetV2 architecture includes first full convolution layer with 32 filters followed by 19 residual bottleneck layers. Links and bottleneck layer are now added on the MobileNet base architecture. The bottleneck residual block is placed between layers. The bottleneck residual block has been developed instead of the deeply separable convolution in the MobileNet architecture. The bottleneck residual block allows the network to calculate activations more efficiently and retain more information after activation. The point-convolutions in the MobileNet architecture keep or increase the number of channels, while

| Type / Stride | Filter Shape | Input Size |
|---|---|---|
| Conv / s2 | $3 \times 3 \times 3 \times 32$ | $224 \times 224 \times 3$ |
| Conv dw / s1 | $3 \times 3 \times 32$ dw | $112 \times 112 \times 32$ |
| Conv / s1 | $1 \times 1 \times 32 \times 64$ | $112 \times 112 \times 32$ |
| Conv dw / s2 | $3 \times 3 \times 64$ dw | $112 \times 112 \times 64$ |
| Conv / s1 | $1 \times 1 \times 64 \times 128$ | $56 \times 56 \times 64$ |
| Conv dw / s1 | $3 \times 3 \times 128$ dw | $56 \times 56 \times 128$ |
| Conv / s1 | $1 \times 1 \times 128 \times 128$ | $56 \times 56 \times 128$ |
| Conv dw / s2 | $3 \times 3 \times 128$ dw | $56 \times 56 \times 128$ |
| Conv / s1 | $1 \times 1 \times 128 \times 256$ | $28 \times 28 \times 128$ |
| Conv dw / s1 | $3 \times 3 \times 256$ dw | $28 \times 28 \times 256$ |
| Conv / s1 | $1 \times 1 \times 256 \times 256$ | $28 \times 28 \times 256$ |
| Conv dw / s2 | $3 \times 3 \times 256$ dw | $28 \times 28 \times 256$ |
| Conv / s1 | $1 \times 1 \times 256 \times 512$ | $14 \times 14 \times 256$ |
| $5\times$   Conv dw / s1 | $3 \times 3 \times 512$ dw | $14 \times 14 \times 512$ |
|      Conv / s1 | $1 \times 1 \times 512 \times 512$ | $14 \times 14 \times 512$ |
| Conv dw / s2 | $3 \times 3 \times 512$ dw | $14 \times 14 \times 512$ |
| Conv / s1 | $1 \times 1 \times 512 \times 1024$ | $7 \times 7 \times 512$ |
| Conv dw / s2 | $3 \times 3 \times 1024$ dw | $7 \times 7 \times 1024$ |
| Conv / s1 | $1 \times 1 \times 1024 \times 1024$ | $7 \times 7 \times 1024$ |
| Avg Pool / s1 | Pool $7 \times 7$ | $7 \times 7 \times 1024$ |
| FC / s1 | $1024 \times 1000$ | $1 \times 1 \times 1024$ |
| Softmax / s1 | Classifier | $1 \times 1 \times 1000$ |

**Figure 3 MobileNet architecture.** 

the bottleneck residual blocks in the MobileNetV2 architecture reduce the number of channels. Since the bottleneck layers are linear, it also prevents non-linear layers from losing too much information (*Sandler et al., 2018*). The general architecture of MobileNetV2 is presented in Fig. 4.

A new scaling model using composite coefficient in the EfficientNet convolutional neural network architecture is proposed. Other convolutional neural networks randomly scale different dimensions such as width, depth, and resolution. In contrast, EfficientNet scales all dimensions equally using a fixed scaling factor. Compound scaling method increased model accuracy and efficiency over traditional scaling methods. The composite scaling method can detect that if the input image is large, more layers are needed, and more channels are needed to detect smaller details in the large image. The EfficientNet architecture basically uses mobile inverted bottleneck convolution. EfficientNetB0 is a revision of the EfficientNet mesh for mobile and embedded devices. The EfficientNetB0 network consists of 5.3 million parameters. Based on the inverted bottleneck residual blocks used in the MobileNetV2 network, EfficientNetB0 adds squeeze-and-excitation blocks (*Tan & Le, 2019*). The EfficientNetB0 architecture is presented in Fig. 5.

| Input | Operator | $t$ | $c$ | $n$ | $s$ |
|---|---|---|---|---|---|
| $224^2 \times 3$ | conv2d | - | 32 | 1 | 2 |
| $112^2 \times 32$ | bottleneck | 1 | 16 | 1 | 1 |
| $112^2 \times 16$ | bottleneck | 6 | 24 | 2 | 2 |
| $56^2 \times 24$ | bottleneck | 6 | 32 | 3 | 2 |
| $28^2 \times 32$ | bottleneck | 6 | 64 | 4 | 2 |
| $14^2 \times 64$ | bottleneck | 6 | 96 | 3 | 1 |
| $14^2 \times 96$ | bottleneck | 6 | 160 | 3 | 2 |
| $7^2 \times 160$ | bottleneck | 6 | 320 | 1 | 1 |
| $7^2 \times 320$ | conv2d 1x1 | - | 1280 | 1 | 1 |
| $7^2 \times 1280$ | avgpool 7x7 | - | - | 1 | - |
| $1 \times 1 \times 1280$ | conv2d 1x1 | - | k | - | |

**Figure 4 MobileNetV2 architecture.**

NASNet enables the creation of the most suitable convolutional neural network architecture using reinforcement learning method. In reinforcement learning, which is used in the NASNet network, the accuracy obtained on the trained dataset of an architecture is used as the result of each search operation. It serves as a reward for the search. Convolutional layers, which generally give good results on different datasets can be used extensively thanks to reinforcement learning. Although the general architecture is defined in the NASNet network, convolutional cells are not certain. The structures of normal and reduction cells in the NASNet network are searched by supervised recurrent neural networks. The number of repetitions of convolutional cells and the number of initial convolution filters in the NASNet network are adjustable parameters. Adjustable parameters are used for scaling. Convolutional cells are of two types, normal and reduction cells. Normal cells are convolutional cells that return a feature map of the same size. A reduction cell is a cell in which the height and width of the feature map are reduced. NASNet Mobile is a revised version of the NASNet network for mobile and embedded devices, reducing the number of parameters. The original NASNet network consisted of 88.9 million parameters, while NASNet Mobile contained 12 cells and 5.6 million parameters (*Saxen et al., 2019*).

Tensorflow is a free and open source software library developed by Google using the C++ programming language used in machine learning and artificial intelligence studies. Tensorflow supports Javascript, C++, Java and Python programming languages. Tensorflow also supports Linux, MacOS, Windows, Android and iOS operating systems (*Tensorflow, 2022*). Tensorflow uses tensors for computation. Tensor is an n dimensional vector/matrix representing data types. The vector is a one-dimensional tensor while the matrix is a two-dimensional tensor. In Tensorflow, all operations take place in graphs. A graph is a set of nodes representing operations in the model, and the connections between

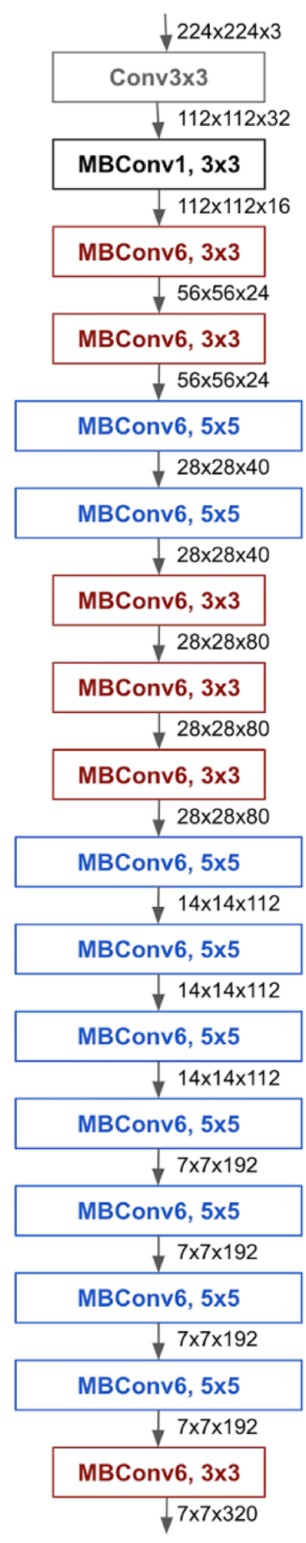

**Figure 5** **EfficientNetB0 architecture.**

them represent the cascading computations. All calculations in the graph are done by connecting tensors together (*Janardhanan, 2020*). Tensorflow can run on central processing unit (CPU), graphics processing unit (GPU), and tensor processing unit (TPU). Tensorboard application is used to visualize the work done using Tensorflow (*Şeker, Diri & Balık, 2017*). Tensorflow library has been preferred in this study because it gives better results in image recognition and classification problems and is widely used in the literature.

Transfer learning is the reuse of a pre-trained machine learning model. Transfer learning is a method that provides short training time and high accuracy performance by reusing the model developed for a task for another application (*Bozinovski & Fulgosi, 1976*). When training a new model with transfer learning, the architecture and weights of the previously trained model are used. The training data and computational power required for transfer learning are less than what was required to train the model from scratch. Transfer learning is widely used in image recognition, speech recognition and natural language processing (*Zhuang et al., 2021*). In this study, pre-trained convolutional neural network models on ImageNet were used to train the fake face detection model.

Ensemble learning is a method of machine learning that relies on combining multiple models to improve the performance of a model. The most popular ensemble learning methods are Bagging, Boosting, and Stacking. The bagging ensemble learning method is to train more than one model with a different sample of the same training dataset. The boosting ensemble learning method involves multiple models making sequential predictions for each sample in the training process. The average of the weights of the predictions made by all models after the prediction made by each model is used by the next model (*Wen & Hughes, 2020*). The stacking ensemble learning method is the training of multiple models separately on the same dataset and combining the predictions of these models during the prediction process. The reason ensemble learning is efficient is that each machine learning model works differently. Each model may perform well on some data and less than others. When different models are combined, they eliminate each other's weaknesses (*Cui et al., 2021*). In this study, the stacking ensemble learning method was used.

## RESULTS AND DISCUSSION

Python 3.6 and Tensorflow software library were used in this study. For training and testing, a dataset containing a total of 140,000 images, 70,000 of which is fake and 70,000 of real faces, was used. 80% of the data set was used for training and 20% for testing. Training was carried out on pre-trained models on ImageNet with the transfer learning method. Lightweight convolutional neural networks are preferred to ensure that the model can operate at maximum performance on mobile and embedded devices. In this study, MobileNet, MobileNetV2, EfficientNetB0, and NASNetMobile convolutional neural networks, which have proven themselves with good results in mobile and embedded devices, were used. In the first stage, these algorithms are used with transfer learning, while the other layers are frozen. A smoothing, a dense layer with ReLU activation function, and a dense layer consisting of two nodes are added. After the models were trained, the EfficientNetB0 convolutional neural network achieved the highest accuracy rate of 93.64%.

```
Layer (type)              Output Shape         Param #
=================================================================
efficientnet-b0 (Functional  (None, 1280)       4049564
)

flatten_5 (Flatten)       (None, 1280)          0

dense_6 (Dense)           (None, 128)           163968

dense_7 (Dense)           (None, 2)             258

=================================================================
Total params: 4,213,790
Trainable params: 4,171,774
Non-trainable params: 42,016
```

**Figure 6 Revised EfficientNetB0 network for transfer learning.**

The change made for transfer learning from the EfficientNetB0 algorithm is shown in Fig. 6.

In order to increase the accuracy, the number of layers added for transfer learning in the EfficientNetB0 algorithm, which has the highest accuracy, has been increased while minimizing the changes to the number of parameters. The EfficientNetB0 convolutional neural network has two dense layers (256 neurons) with ReLU activation function, two dropout layers, one flatten layer, one dense layer (128 neurons) with ReLU activation function, and a dense layer with two nodes and softmax activation function used for classification. These layers are added while the previous layers are frozen during transfer learning. As a result of this process, the number of parameters is 4,476,446. The accuracy rate achieved with the revised model is 95.48%. The new change for transfer learning from the EfficientNetB0 algorithm is shown in Fig. 7.

In order to improve the success rate achieved with the new EfficientNetB0 model, the ensemble learning method, which has been used extensively recently, has been used. In this case, the accuracy was 96.20%. Then EfficientNetB0 convolutional neural network achieved the best results with the ensemble learning method called stacking, while the MobileNetV2 model, which had the highest accuracy, were trained together. Then, the MobileNet model was added, and the accuracy rate was 96.41% when the three models were used with the stacking ensemble learning method. Finally, the NASNetMobile model was added, and the accuracy rate was 96.27% when the four models were used with the stacking ensemble learning method. The results of all trials are shown in Table 3.

As can be seen in Table 3, when the revised EfficientNetB0, MobileNet, and MobileNetV2 models were reused with the stacking ensemble learning model, they achieved the highest accuracy rate of 96.44%. Tests conducted on the Samsung Galaxy A24 device (Octa-core 2 × 2.2 GHz Cortex-A76 & 6 × 2.0 GHz Cortex-A55 CPU, 6 GB RAM) showed that the proposed model operated with a latency of 0.171 s.

The CelebA-HQ dataset was used to assess the impact of the proposed method on other datasets (*Huang et al., 2018*). CelebA-HQ dataset is a high-quality version of CelebA-HQ

```
Layer (type)              Output Shape          Param #
=================================================================
efficientnet-b0 (Functional  (None, 1280)          4049564
)

dense_20 (Dense)          (None, 256)            327936

activation_4 (Activation) (None, 256)            0

dropout_8 (Dropout)       (None, 256)            0

dense_21 (Dense)          (None, 256)            65792

dropout_9 (Dropout)       (None, 256)            0

flatten_9 (Flatten)       (None, 256)            0

dense_22 (Dense)          (None, 128)            32896

dense_23 (Dense)          (None, 2)              258

=================================================================
Total params: 4,476,446
Trainable params: 4,434,430
Non-trainable params: 42,016
```

**Figure 7 New revision for transfer learning from EfficientNetB0 algorithm.**

**Table 3 Performance metrics of fake face image detection models.**

| Algorithm | Accuracy (%) | Precision (%) | Recall (%) | F1 Score (%) | Parameter number |
|---|---|---|---|---|---|
| Revised EfficientNetB0+MobileNetV2+MobileNet | 96.44 | 97.82 | 97.36 | 97.58 | – |
| Revised EfficientNetB0+MobileNetV2+MobileNet+NASNetMobile | 96.27 | 96.25 | 96.27 | 96.25 | – |
| Revised EfficientNetB0+MobileNetV2 | 96.20 | 97.59 | 96.07 | 96.83 | – |
| Revised EfficientNetB0 | 95.48 | 96.14 | 95.43 | 95.78 | 4,476,446 |
| EfficientNetB0 | 93.64 | 93.70 | 93.27 | 93.48 | 4,213,790 |
| MobileNetV2 | 91.12 | 92.53 | 91.87 | 92.19 | 3,500,000 |
| MobileNet | 87.83 | 93.66 | 87.25 | 90.34 | 4,253,864 |
| NASNetMobile | 78.87 | 79.12 | 78.04 | 78.57 | 5,600,000 |

consisting of 30,000 images (*Odhiambo, 2021*). A total of 30,000 fake images generated using StyleSwin were also utilized. StyleSwin is a transformer-based Generative Adversarial Network (GAN) used for generating high-resolution images (*Zhang et al., 2022*). The dataset comprises a total of 60,000 images, including 30,000 fake and 30,000 real images. 80% of the dataset was used for training, and the remaining 20% for testing. The hyperparameters in the proposed method remained unchanged for training. The comparison of the results obtained with the proposed method on CelebA-HQ and FFHQ datasets is shown in Table 4.

**Table 4 The comparison of the results obtained with the proposed method on CelebA-HQ and FFHQ datasets.**

| Algorithm | Dataset | Accuracy (%) | Precision (%) | Recall (%) | F1 Score (%) |
|---|---|---|---|---|---|
| Revised EfficientNetB0+MobileNetV2+MobileNet | FFHQ | 96.44 | 97.82 | 97.36 | 97.58 |
| Revised EfficientNetB0+MobileNetV2+MobileNet | CelebA-HQ | 94.52 | 94.67 | 94.29 | 94.47 |

**Table 5 Comparison of the proposed model with previous studies.**

| Study | Method | Algorithm | Feature | Dataset | Performance metric | Accuracy rate |
|---|---|---|---|---|---|---|
| Proposed model | Deep learning | Revised EfficientNetB0+MobileNetV2+MobileNet | Fake face detection | FFHQ and StyleGAN2 | Accuracy | 96.44% |
| Wang et al. (2021) | Deep learning | New model based on neuron monitoring | Fake face detection | FFHQ and StyleGAN2 | Accuracy | 91.9% |
| Suganthi et al. (2022) | Deep learning | FisherFace | Fake face detection | FFHQ | Accuracy | 94.92% |
| Bang & Woo (2021) | Deep learning | MobileNetV3 | Fake face detection | FFHQ and StyleGAN2 | Accuracy | 83.64% |
| Hu, Li & Lyu (2020) | Image processing | Dlib library | Fake face detection | FFHQ and StyleGAN2 | AUC | 94% |
| Guo et al. (2022) | Deep learning | EfficientNet-B5 | Fake face detection | FFHQ and StyleGAN2 | AUC | 91% |

As seen in Table 4, the proposed method achieved an accuracy rate of 94.52% and a latency of 0.170 s on the CelebA-HQ dataset, demonstrating good results. Comparison of the proposed model with previous studies is shown in Table 5.

As can be seen in Table 5, the proposed model has reached a higher accuracy rate than previous studies thanks to the use of transfer learning and ensemble learning methods.

## CONCLUSIONS

Fake face images can be used for machine learning, image processing, *etc.*, are content produced using techniques. The most common type of digital manipulation today. The use of machine learning algorithms in fake face generation makes it increasingly difficult to distinguish content from real images. In this study, MobileNet, MobileNetV2, EfficientNetB0, and NASNetMobile lightweight convolutional neural networks were used for the detection of fake face images. Due to the widespread use of mobile devices, lightweight convolutional neural networks that can work on mobile devices were preferred in this study. A dataset containing 70,000 real and 70,000 fake images was used for the training process. EfficientNetB0 algorithm has reached the highest accuracy rate as a result of the training processes. In order to increase the accuracy rate, the highest accuracy rate of 96.44% was achieved when EfficientNetB0, MobileNet and MobileNetV2 were used together with the ensemble learning method. Applying filters to real images, unnatural shapes on faces, heavy make-up, glasses, metal, *etc.*, on the face presence of accessories and child images cause the model to make inaccurate inferences. In this study, the dataset lacks

sufficient data in some categories such as accessories, shapes, makeup, and young age on faces. Additionally, it is unknown which country 85% of the facial images in the dataset belong to. There may be very few or no images from some countries in the dataset. This situation negatively affects the generalizability of the model. Also, for lightweight convolutional neural networks to work with high accuracy, hyperparameter configuration and a considerable amount of data are required. Fake face detection systems may restrict user privacy as they may enable the analysis of personal private images. In this study, publicly available image datasets were used for model training and testing processes. Authorities may find fake face detection systems restrictive to freedom of expression due to the potential for teaching non-fake content as fake. However, if fake face detection systems are managed by independent and reliable organizations, the risks of inhibiting freedom of expression can be mitigated. The study only focuses on detecting fake face images. In subsequent studies, the method will be expanded to enable the detection of manipulated text, audio, or video content. Increasing the diversity of the data set will significantly increase the success of the model in order to prevent situations where the model makes wrong predictions in future studies. In addition, in future studies identification change, attribute manipulation and expression manipulation detection can be added to the model with transfer learning, so that four basic face manipulations can be detected.

### Funding
The authors received no funding for this work.

### Competing Interests
Emre Şafak is employed by HAVELSAN.

### Author Contributions
- Emre Şafak conceived and designed the experiments, performed the experiments, analyzed the data, performed the computation work, prepared figures and/or tables, authored or reviewed drafts of the article, and approved the final draft.
- Necaattin Barışçı conceived and designed the experiments, analyzed the data, authored or reviewed drafts of the article, and approved the final draft.

### Data Availability
The Flickr-Faces-HQ Dataset (FFHQ) is available at GitHub: https://github.com/NVlabs/ffhq-dataset.

StyleGAN2 is available at GitHub: https://github.com/NVlabs/stylegan2.

The CelebA-HQ celebrity faces dataset is available at Kaggle and Zenodo:

- https://www.kaggle.com/datasets/badasstechie/celebahq-resized-256x256/data.

- Şafak, E. (2024). CelebA-HQ Dataset Generated Images with StyleSwin Method. Zenodo. https://doi.org/10.5281/zenodo.11118066.

The code is available in the Supplemental Files.

## Supplemental Information

Supplemental information for this article can be found online at http://dx.doi.org/10.7717/peerj-cs.2103#supplemental-information.

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
