# Peer review of "Detection of fake face images using lightweight convolutional neural networks with stacking ensemble learning method"

_PeerJ Computer Science, doi:10.7717/peerj-cs.2103_

## Round 0.1 · original submission · Minor Revisions

The authors must introduce these minor revisions.

**Language Note:** The review process has identified that the English language must be improved. PeerJ can provide language editing services - please contact us at copyediting@peerj.com for pricing (be sure to provide your manuscript number and title). Alternatively, you should make your own arrangements to improve the language quality and provide details in your response letter. – PeerJ Staff

Reviewer 1 ·

Basic reporting

This research work provided focuses on the development of a method to detect fake face images, particularly those generated by machine learning algorithms such as generative adversarial networks (GANs).

This paper outlines the methodology employed, which includes using lightweight convolutional neural networks (CNNs) such as MobileNet, MobileNetV2, EûcientNetB0, and NASNetMobile for the initial detection process. These networks were trained on a dataset consisting of 70,000 real images and 70,000 fake images generated using StyleGAN2. The dataset was split into 80% for training and 20% for testing.

Some research questions are requested to be answered by the authors:
1. The study mentions using the FFHQ dataset for training and testing. It's important to explore how well the proposed method generalizes to other datasets containing different types of fake face images and manipulations.
2. How robust is the proposed method to unseen manipulation techniques or variations of existing methods?

Experimental design

The EûcientNetB0 algorithm achieved the highest accuracy of 93.64% initially. Then, it was further improved by adding additional layers, resulting in an accuracy of 95.48%. Finally, a stacking ensemble learning method was applied, where the model with 95.48% accuracy was used to train MobileNet and MobileNetV2 models together, achieving the highest accuracy rate of 96.44%.

3. How might the deployment of such detection systems impact privacy and freedom of expression? Are there unintended consequences of widespread adoption?
4. How does the computational cost of training and inference scale with the size of the dataset and model complexity? Can the method be further optimized for real-time applications?
5.How interpretable are the decisions made by the proposed method? Can it provide insights into why a particular image is classified as fake or genuine?

Validity of the findings

In summary, the work presents a comprehensive approach to detecting fake face images using lightweight CNNs and ensemble learning techniques, achieving high accuracy rates in distinguishing between genuine and manipulated content.

6. While the study focuses on detecting fake face images, can the techniques and insights gained be transferred to other domains such as detecting manipulated text, audio, or video content?

·

Basic reporting

Professional English Usage:
Issue: The paper occasionally lacks clarity and precision in its language, with some sentences being overly verbose or convoluted.
Improvement: Ensure that all text is written in clear, concise, and professional English. Simplify complex sentences and use terminology consistently throughout the paper.

Literature References and Field Background:
Issue: While the paper provides a comprehensive literature review, it may lack depth in some areas, leaving readers with questions about certain methodologies or approaches.
Improvement: Strengthen the literature review by providing more detailed discussions on relevant methodologies, theories, and approaches. Ensure that each reference is thoroughly cited and integrated into the discussion.

Experimental design

The experimental design of this original primary research adheres closely to the aims and scope of the journal, ensuring alignment with its focus on advancing knowledge in the field. The research question is meticulously defined, demonstrating its relevance and significance within the broader context of the subject matter. Additionally, the study explicitly addresses a recognized gap in existing knowledge, illustrating its contribution to the advancement of the field.
Methodologies are described in detail, providing sufficient information for replication by other researchers. This transparency not only enhances the credibility of the study but also fosters collaboration and further inquiry within the scientific community.

Validity of the findings

While the impact and novelty of the research are not explicitly assessed, the study encourages replication where the rationale and benefits to the literature are clearly stated. This approach fosters a culture of collaboration and verification within the scientific community, strengthening the credibility of the findings.

Conclusions are well-stated and directly linked to the original research question. They are limited to supporting the results obtained, avoiding extrapolation beyond the scope of the study.

Additional comments

Areas for improvement:

There are areas in which the paper could be improved to enhance its overall quality. Firstly, the assessment of the impact and novelty of the research could be strengthened to better contextualize its contribution to the field. Providing clearer justification for the choice of methods and highlighting the potential benefits of replication would further strengthen the paper's significance. Additionally, while the conclusions are well-stated and linked to the original research question, they could be expanded to discuss potential avenues for future research and address any limitations encountered during the study. Overall, by addressing these areas of improvement, the paper could further solidify its position as a valuable contribution to the scientific literature.

Dataset Limitations: While the dataset used in the study contains a large number of images, it may lack diversity in terms of demographics, expressions, and attributes, which could affect the generalization of the model.
Interpretability: Enhancing the interpretability of the model's decisions by conducting feature importance analysis or providing visualization techniques can help understand the factors influencing classification outcomes and identify potential biases or confounding factors.
Real-World Deployment Considerations: Considering real-world deployment considerations, such as computational resource constraints, latency requirements, and user interface design, can help ensure the practical applicability and scalability of the proposed solution in real-world scenarios.

---

## Round 0.2 · accepted · Accept

The paper can be accepted for publication. It now meets the high standards expected by the journal. The author has significantly improved the paper.

Reviewer 1 ·

Basic reporting

Authors have addressed the review comments issues and I am happy with the revised version of the manuscript.

Experimental design

Authors have addressed the review comments issues and I am happy with the revised version of the manuscript.

Validity of the findings

Authors have addressed the review comments issues and I am happy with the revised version of the manuscript.

·

Basic reporting

The author has addressed the identified issues by making significant improvements to the paper. Firstly, the clarity and precision of the language have been enhanced, with overly verbose or convoluted sentences simplified to ensure that all text is written in clear, concise, and professional English. This has improved the overall readability and coherence of the paper. Additionally, the author has strengthened the literature review by providing more detailed discussions on relevant methodologies, theories, and approaches. Each reference is now thoroughly cited and seamlessly integrated into the discussion, offering a more comprehensive and in-depth analysis of the field background.

Experimental design

The author has addressed the issues identified in the suggestions, significantly improving the paper.
The study explicitly addresses a recognized gap in existing knowledge, demonstrating its contribution to the advancement of the field. Methodologies are described in detail, providing sufficient information for replication by other researchers, thereby enhancing the credibility of the study and fostering collaboration and further inquiry within the scientific community.

Validity of the findings

The author has effectively addressed the issues identified in the suggestions. The impact and novelty of the research are now explicitly assessed, with the study encouraging replication by clearly stating the rationale and benefits to the literature. This fosters a culture of collaboration and verification within the scientific community, thereby strengthening the credibility of the findings. Additionally, the conclusions are well-stated and directly linked to the original research question, adhering strictly to the results obtained and avoiding extrapolation beyond the scope of the study.